# Study on the Discoloration Mechanism of Eucalyptus Wood during Thermal Treatment in Different Media

**DOI:** 10.3390/polym15071599

**Published:** 2023-03-23

**Authors:** Peng Zhang, Jianmin Gao, Fu Liu, Yao Chen, Yao Peng

**Affiliations:** 1Research Institute of Wood Industry, Chinese Academy of Forestry, Beijing 100091, China; 2MOE Key Laboratory of Wooden Material Science and Application, College of Materials Science and Technology, Beijing Forestry University, Beijing 100083, China

**Keywords:** chromophore structure, thermal treatment, discoloration mechanism, color regulation

## Abstract

Chromophore structures in wood are the core elements for regulating wood color. Thermal treatment can regulate the color of wood, thus increasing its added value. In this study, conventional thermal treatment was used to regulate the color of Eucalyptus, in order to make its color close to the precious wood species Burma padauk. The color change in Eucalyptus wood was analyzed using the chromaticity index and UV–Vis. The chromophore structures in the treated wood and their discoloration mechanisms were characterized via FTIR, XPS, NMR, etc. The results showed that the color of eucalyptus could be regulated via thermal treatment to become more similar to the color of Burma padauk under both saturated steam and hot air. The treated wood showed a color difference in the 400~500 nm region in spectral absorption. The changes in the chromophore structures of wood were accompanied by the degradation of hemicelluloses. Meanwhile, demethoxylation occurred in the syringyl structure G of lignin, which led to the polymerization of lignin and decreased the lightness value of wood. Moreover, the number of conjugated structures in the chromophore groups increased, which caused the color of the wood to tend toward red. This study provides a reference for the color regulation of wood, and the mechanisms are also discussed.

## 1. Introduction

As a natural polymer, wood is the most widely used biomass material. The processing technology of wood has continuously advanced toward low cost, high value, and environmental preservation with the increase in demand [1,2]. Fast-growing wood is the main species used in the Chinese wood industry. It generally has defects such as low density, low mechanical strength, and uneven color. Due to the large-scale use of fast-growing forests in China, the color of the wood is an important feature that needs to be improved urgently. Various modification processes such as densification [3,4], dyeing [5], and recombination [6] have been carried out for the modification of fast-growing wood, among which wood heat treatment is considered to be a green and environmentally friendly wood modification method. Heat treatment can adjust the moisture content of wood and improve dimensional stability, corrosion resistance, etc. [7]. It is a routine process widely used in the wood industry. Wood color turns into dark red after thermal treatment, and the color tends to be uniform and stable [8]. It is important to realize the high-value and low-cost modification of fast-growing wood by means of heat treatment, which is recognized as a conventional treatment process to improve the color aesthetics and uniformity of wood.

It has been reported that there is a correlation between the heat-treatment process and the degree of color change [9,10,11]. The processing factors that affect wood color include temperature, relative humidity, pressure, treatment time, moisture content, pH value, etc. Temperature is the main influencing factor. The degree of discoloration is positively correlated with the treatment temperature [12]. For high-pressure steam-treated wood, the color difference increases with the increase in treatment pressure and duration [13]. Wood with a high moisture content is more prone to discoloration than dry wood. The presence of moisture promotes a decrease in the softening temperature of lignin, lowering the minimum temperature required for discoloration. The influence of the water in the cell wall is much greater than the water in the cell lumen and intercellular space [14]. The pH value in the thermal system also plays an important role in the wood thermal discoloration process. For example, pterostilbene is light yellow in an acidic environment with a pH value of 5.5 and turns dark brown in an alkaline environment with a pH value of 8.7 [15]. During heat treatment, polysaccharides are degraded, which results in a decrease in acetyl groups and alcoholic hydroxyl groups, the degradation of acetyl groups generating acetic acid, and leading to a decrease in the pH value of the wood system [16]. An acid environment catalyzes the degradation of lignin [17] and promotes the occurrence of color reactions, which intensifies the change in wood color. Oxidation or condensation reactions induced by heat treatment can promote the formation of a color system in wood similar to that of precious tree species, which can adjust the color of fast-growing tree species and increase their use value.

Wood contains a variety of structural systems, such as skeleton structures composed of cellulose and hemicelluloses with ordered and disordered arrangements of polysaccharide molecules; filling structures composed of lignin, which are formed by the interconnection of benzene ring macromolecules; and chromophore structures in lignin and extractives. These compositions together affect the wood color. Chromophore structures are the key factors in the regulation of wood color. Wood components undergo changes in structural properties when exposed to heat. In addition to the degradation of polysaccharides, lignin also undergoes significant changes [18]. During heat treatment, β-O-4 on lignin undergoes cleavage and demethoxylation reactions [19], and condensation reactions occur in lignin macromolecules [14,20], resulting in a decrease in the methoxyl, hydroxyl, and guaiacol unit content [21,22]. The conjugated carbonyl content increases, forming p-quinone and hydroquinone structures [23] and producing acetic acid and phenolic compounds [24], which result in changes in wood color [25,26]. The structure of the extractives is unstable under heat treatment, and this easily causes color changes during heat treatment. The extractives undergo oxidation and hydrolysis reactions to form colored substances [27], which is the main reason for wood discoloration. The hydroxyl-OH in the extractives is oxidized to form a conjugated chromogenic structure that causes the wood to turn red [28]. Flavan 3,4-diol is oxidized to form a brown-red substance [29]. Condensed tannins can produce an insoluble reddish-brown complex when heated in an acidic environment [30]. Hemicellulose degrades during heat treatment, which leads to an increase in the colored composition [31].

In order to improve the added value of fast-growing wood, in this study, a color control process was carried out using a conventional heat-treatment process. The influence of the main parameters, temperature, and pressure on the color of thermally treated wood was investigated. The chromophore system was taken as the research object under different heat processes. Microscopic characterization methods were used to explain the relationship between the chromophore and the wood color. In the process of wood color control, the target tree species for wood color adjustment was determined to provide a reference for realizing the industrial utilization of wood color control.

In this study, we selected the fast-growing tree species Eucalyptus (*Eucalyptus grandis × urophylla*) as the control species and used the main precious tree species Burma padauk (*Pterocarpus macrocarpus*) as the target tree species to compare the color and chromophore system differences after thermal treatment in hot air and saturated steam. By means of FTIR, UV, NMR, and XPS, the adjustability of the chromophore system was investigated. The wood color change under different heat-treatment conditions (medium, temperature, and treatment time) was compared to find the best process. The aim of this study was to investigate the color change process of Eucalyptus wood (fast-growing species) that would turn it into the color of Burma padauk (*Pterocarpus macrocarpus*), i.e., upgrading the fast-growing species to a precious tree species by changing its color, thus increasing the value of timber.

## 2. Materials and Methods

### 2.1. Wood Samples

Eucalyptus wood (*Eucalyptus grandis × urophylla*) samples were collected from Guangxi Province, southwest China. The tree was 5 years old, with a moisture content of 100~110%. Wood logs were cut into discs and 3 discs from different heights from the trunk were chosen cut to pieces and ground into powder, and portions between 250 and 425 μm were collected. Then, the powder was oven-dried to a moisture content of 7~12%. According to ASTM D1105-1996, the extractives in the powder were removed via an extraction process with benzene/alcohol (*v*/*v*, 2/1) in a Soxhlet extractor for 24 h. After extraction, the wood powder was dried for 2 h at 103 °C and then put in a desiccator for further treatment.

Burma padauk (*Pterocarpus macrocarpus*) samples were obtained from the Wood Drying and Functional Improvement Laboratory of the School of Materials Science, Beijing Forestry University. The wood pieces were crushed and the powder between 250 and 425 µm was collected.

### 2.2. Thermal Treatment

Thermal treatments were performed in hot air and saturated steam, respectively. Briefly, 10 samples for each treatment were used. Each sample was 1.0 g of wood flour spread out in a glass dish. The wood powder was used for heat treatment due to the following considerations: (1) to avoid uneven heat caused by heat transfer during treatment; (2) to meet the requirements of instrument testing for test pieces (FTIR, NMR, and XPS); and (3) to focus on changes in the wood’s chemical structure, ignoring the impact of wood structure.

For hot-air treatment, the samples were heated in an electric blast drying oven at 180 °C for 12 h. Before heat treatment, the oven was preheated to the specified temperature. After thermal treatment, the wood powder was left for 2 h and measured after the moisture content was stable.

Saturated-steam treatment was performed in a pressure sterilizer (Binjiang Medical LS-35HD, Jiangyin, China) at 130 °C for 12 h. The pressure was controlled at 0.27 MPa. Before heat treatment, distilled water was added to ensure the generation of saturated steam. A glass dish with a diameter of 15 mm was selected to ensure that the wood powder is in contact with the saturated steam as large as possible. For each glass dish, 3~5 g of wood powder was added for treatment. The glass dish was placed on the bottom of the container. After thermal treatment, the wood powder was left for 2 h and measured after the moisture content was stable. 

### 2.3. Color Measurement

The colorimetric data were collected in a DF110 spectrophotometer (Konica Minolta CM-2300d, Tokyo, Japan) using a 10° standard observer and standard illuminant D65 with an 8 mm aperture according to the CIE*L***a***b** system. Color measurements were performed at three points (five measures for each point), and the average value was calculated. The color coordinates lightness *L** (varying from 0 for black to 100 for white), *a** (varying from negative values for green to positive values for red on the red–green axis), and *b** (varying from negative values for blue to positive values for yellow on the yellow–blue axis) were measured. Differences in the color of the samples before and after heat treatment were determined based on the following Equations (1)–(4):Δ*L*^*^ = *L*^*^ − *L*^*^_0_(1)
Δ*a*^*^ = *a*^*^ − *a*^*^_0_(2)
Δ*b*^*^ = *b*^*^ − *b*^*^_0_(3)
(4)$$\Delta E*=\sqrt[{2}]{\Delta L^{*2}+\Delta a^{*2}+\Delta b^{*2}}$$
where Δ*L**, Δ*a**, Δ*b**, and Δ*E** represent the variations in the lightness, green–red coordinate, blue–yellow coordinate, and total color difference, respectively. 

### 2.4. UV–Vis Spectrum

DRUV–Vis (diffuse reflectance ultraviolet–visible spectroscopy) was recorded over the wavelength range of 200~700 nm on a UV–Vis spectrophotometer (Shimadzu UV-2550, Kyoto, Japan) equipped with an integrated sphere. BaSO_4_ was used as a reference sample. The wood samples were prepared as thin slices (30 mm diameter) using EVA (ethylene-vinyl acetate copolymer) hot glue so that the loose powder could be detected in the spectrophotometer.

### 2.5. FTIR Analysis

FTIR (Fourier transform infrared spectroscopy) spectra were recorded in absorbance mode using a PERKIN Elmer Spectrum Gx instrument (Perkin Elmer, Shanghai, China). The wood powder over 200 mesh was sieved to mix with KBr powder. The concentration of the samples in a KBr pellet was about 1%. The number of scans was 32, the resolution was 4 cm^−1^, and the sweep scan range was 400~4000 cm^−1^.

### 2.6. NMR Spectrum

For ^13^C CP-MAS NMR (^13^C cross-polarization magic-angle spinning nuclear magnetic resonance) spectroscopy, a JNM-ECZ600R NMR spectrometer (JEOL, Tokyo, Japan) was used with a magnetic flux density of 14.09 T. Carbon spectra were acquired with a 4 mm bore probe head. The spinning rate was set to 15 kHz. In total, 1024 scans were collected with a 3 s delay between successive scans. The length of the contact time was 2 ms, and the spectral width was 37.89 kHz.

### 2.7. XPS Spectrum

XPS (X-ray photoelectron spectroscopy) analysis was performed on the samples (<74 μm) with a ThermoVG Scientific Sigma Probe (Thermo Fisher ESCALAB 250Xi, Waltham, MA, USA) using a microfocusing monochromatic AlKa X-ray source at an operating pressure between 10^–9^ and 10^–8^ mbar. A high-resolution scan was conducted on the C_1S_ peak from 280 to 300 eV and the O_1S_ peak from 530 to 535 eV for each sample. The chemical bond analysis of carbon was performed by studying the fit of the C_1S_ peak and deconvoluting it into four subpeaks for C-C/C=C, C-O, and C=O- groups, with areas represented by C_1_, C_2_, and C_3_, respectively. The O_1S_ signals were similarly deconvoluted into two subpeaks for C=O and C-O groups and represented by O_1_ and O_2_, respectively. The ratio of C_3_/C_2_ and O_1_/O_2_ was calculated.

## 3. Results and Discussion

### 3.1. Color Change and UV–Vis Analysis

Figure 1 shows the pictures and color parameters of Eucalyptus wood before and after treatment. After heat treatment, the value of *a** in Eucalyptus wood increased by 50%, which was close to the *a** value of the target wood. The *L** value of the saturated-steam-treated wood (38.74) at 130 °C was closer to the target wood (39.61). The *b** value was less affected by heat treatment, as it changed slightly after treatment. For the treatment at 180 °C using hot air, the *L** value decreased by more than 50%, which led to a darker appearance and a higher Δ*E** value. It can be seen that both saturated steam and hot air successfully regulated the wood color, which turned into a dark red color.

As shown in Figure 2, the UV–Vis spectra of the treated samples showed more absorption than the untreated samples and were close to the target wood. The two absorption peaks at 405 and 515 nm did not reach the absorption intensity of Burma padauk [32,33]. The corresponding color parameter was *a**_max_ = 11, which was observed in the spectral range scope of Burma padauk. In peaks above 400 nm, the absorbance of saturated-steam treatment at 130 °C was always higher than that of hot-air treatment, and the *L** value was lower than that of the hot-air treatment.

### 3.2. FTIR Analysis

The structural changes at the molecular level that caused the discoloration are explored and discussed in this section. Figure 3 compares the infrared spectra of the wood treated with the two media and the target wood. The infrared spectra were normalized at 1514 cm^−1^ (benzene ring C-H-stretching vibration), and the relative peak height changes in six main functional groups relative to the benzene-ring peak are listed in Table 1.

Both Burma padauk and 130 °C treated samples showed a narrow peak near 3431 cm^−1^, while the absorption peaks of 180 °C treated samples were wider, indicating more types of hydroxyl groups (such as phenolic hydroxyl, ester hydroxyl, etc.) generated after heat treatment at a higher temperature. In the characteristic absorption region of carbonyl, the content of non-conjugated carbonyl did not significantly change under severe heat conditions, but the relative absorption peak of the 180 °C group was slightly enhanced, and the *b** value of the 180 °C group was higher than that of the other three groups of wood. It was verified that the non-conjugated carbonyl group was positively correlated with the *b** values [34]. The relative content of the conjugated carbonyl group was in the following order: Burma padauk > 130 °C > Eucalyptus wood > 180 °C.

The absorption peaks at 1664 cm^−1^ and 1625 cm^−1^ were related to the *a** value. The absorption peak of the carbonyl group conjugated with the benzene ring at 1664 cm^−1^ is generally generated due to the quinone structure or the carbonyl group conjugated with the aromatic ring (such as α-C=O). During hot-air treatment at 180 °C, the conjugated carbonyl structure broke, so its absorption decreased more obviously than in the untreated samples. However, under saturated-steam treatment, the conjugated carbonyl content increased, and the peak position moved to 1625 cm^−1^ and showed stronger absorption. This is attributed to the α-β-conjugated double bond and carbonyl group in the flavonoids in wood. After the saturated-steam treatment, the intensity of the peak at 1625 cm^−1^ tended to be consistent with the target wood [35].

The main difference between the 130 °C group and Burma padauk was also reflected in the peak near 1270 cm^−1^. In the spectra of Burma padauk, the peak at 1270 cm^−1^ was the C-O-stretching vibration of the methoxy group in the guaiac-based G structure, which mainly reflected the G-type lignin. The number of this structure was more in the system, while this peak intensity was lower in the 130 °C group than in the other three groups of woods. The lignin system with a high content of G-type lignin underwent a large degree of conjugation and polymerization [36,37], while the proportion of G-type lignin structure was less in the treated wood, which explained its color difference with Burma padauk.

### 3.3. XPS Analysis

The number and ratio of conjugated structures in the wood have a significant effect on color. Electron-energy spectrum analysis of carbon atoms and oxygen atoms was used to explore different structures of carbon-oxygen bonds in wood systems. Figure 4 shows the XPS spectra of C1s and O1s of Eucalyptus wood before and after treatment. The electron-binding energies of C1, C2, C3, and C4, the proportion of C atoms, the ratio of C3/C2, the electron-binding energies of O1 and O2, and the ratio of O1/O2 are shown in Table 2.

It can be seen from Figure 4 and Table 2 that the proportion of C1 in Eucalyptus wood increased after heat treatment, and the proportion of C2 changed slightly. The value of C3 and the C3/C2 all decreased, which suggested a decrease in the C=O group content. From the decreasing trend of C4, it can be speculated that part of C3 resulted from the decrease in C=O in the ester group. The value of the carbonyl content showed an increasing trend, as can be seen from the increase in O1 and O1/O2 values. Overall, the heat-treatment process increased the number of carbonyl groups in Eucalyptus wood and reduced the number of ester groups [38,39].

Comparing the saturated-steam treatment and hot-air treatment, the saturated-steam process caused more significant damage to the carboxyl functional group, resulting in a decrease in the number of oxygen-containing groups after treatment. In terms of carbonyl content, there was no significant difference in carbonyl content between the two treatments, and the ratio of O1/O2 in the system was basically the same.

### 3.4. CPMAS 13C-NMR Analysis

Nuclear magnetic resonance spectroscopy can clearly characterize the changes in the chemical structure of wood. Figure 5 shows a comparison of the nuclear magnetic resonance spectra of Eucalyptus wood and Burma padauk wood after the optimal process.

The peak of the hemicelluloses acetyl group in the untreated Eucalyptus was the highest and gradually decreased with the decrease in treatment temperature, indicating that hemicelluloses were degraded during the heat treatment. The absorption of 105 ppm cellulose C1 was relatively stable, and it is generally used as a reference peak to compare the changes in other chemical components [40]. In this study, the semi-quantitative method was used to compare and analyze the differences in the content of the analyzed groups. The cellulose C1 peak was used as the integrated reference area, and the integral value in the range of ∫ (110–95 ppm) was defined as the reference value = 1. The integrated area of the other peaks was divided by the reference value to obtain the content of this group relative to the cellulose C1 peak. Based on the values reported in previous studies [40,41,42], the chemical shifts, the relative integral values of peak areas, and attributions of NMR spectra are listed in Table 3. The content of the basic structural units of lignin changed obviously. The untreated Eucalyptus wood only contained an S structure in lignin. However, after heat treatment, the absorption peak of the G unit (149.25) appeared and strengthened, which reached 50% in content compared with Burma padauk. Meanwhile, the S unit of the treated wood was reduced by 50% after heat treatment. This process was caused by a reaction affecting the methoxyl group (i.e., the reduction in the methoxyl group content at 56 ppm), which indicated that the methoxyl group was seriously desorbed during the treatment at 180 °C, while it slightly reduced after the treatment at 130 °C. The absorption peak at the C1 position (135 ppm) of lignin disappeared after being treated with hot air at 180 °C and slightly decreased after being treated with saturated steam. The peak at the C6 position at 116 ppm did not exist in Eucalyptus, while it appeared and increased in the treated wood. Differences in cellulose crystallinity could also be observed. The crystallinity of the treated wood was similar to that of Burma padauk after saturated-steam treatment at 130 °C. Moreover, the absorption peak in the amorphous region of cellulose reduced, generally lower than that of the target wood.

## 4. Conclusions

In this study, both saturated steam and hot air were used as heat-treatment media to regulate the color of Eucalyptus wood, in order to turn its color into a more similar color to Burma padauk. After heat treatment, the lignin in Eucalyptus wood underwent demethoxylation and polymerization, and the degree of polymerization and the conjugated groups increased, especially for the conjugated C=O structure. The carbonyl content was positively correlated with the *a** value. After the saturated-steam treatment, the conjugated carbonyl structure increased, which made the wood color turn red. Compared with hot-air treatment, the damage degree of carboxyl functional groups was more severe after saturated-steam treatment, resulting in a decrease in the content of oxygen-containing groups. The syringyl (S) structure units in lignin underwent a demethoxylation reaction, which led to the polymerization of lignin. This change in wood increased its absorption of visible light, thus decreasing the *L** value.

Our investigation of different heat-treatment conditions revealed that both pressure and temperature were important heat-treatment conditions that aggravated the color change. Through the analysis of the color change caused by changes in the chemical structure, it was proved that color was a tunable wood parameter under high-temperature and high-pressure thermal treatment. This study provides a theoretical reference for wood thermal discoloration. The experiments proved that the color of fast-growing plantation Eucalyptus wood could be regulated under thermal treatment, and the color of the treated wood could be close to the color of Burma padauk. This research explored a new path for wood surface color regulation in the current heat-treatment process of panels or veneers. Heat treatment cannot fully meet the goal of color regulation, and it needs to be combined with other chemical reactions to realize the regulation of the chromophore system.

## Figures and Tables

**Figure 1 polymers-15-01599-f001:**
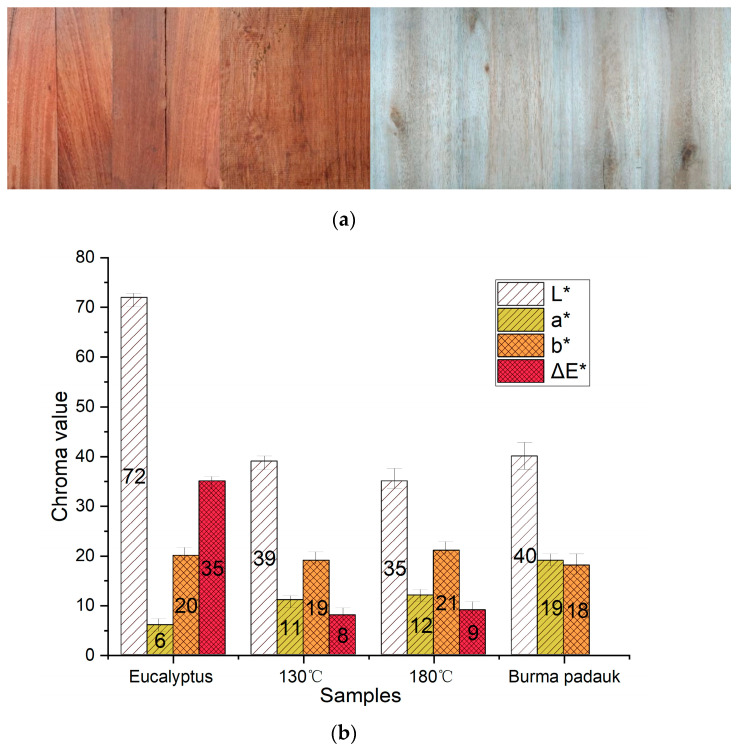
(**a**) Pictures and (**b**) chroma value of Eucalyptus and Burma padauk wood.

**Figure 2 polymers-15-01599-f002:**
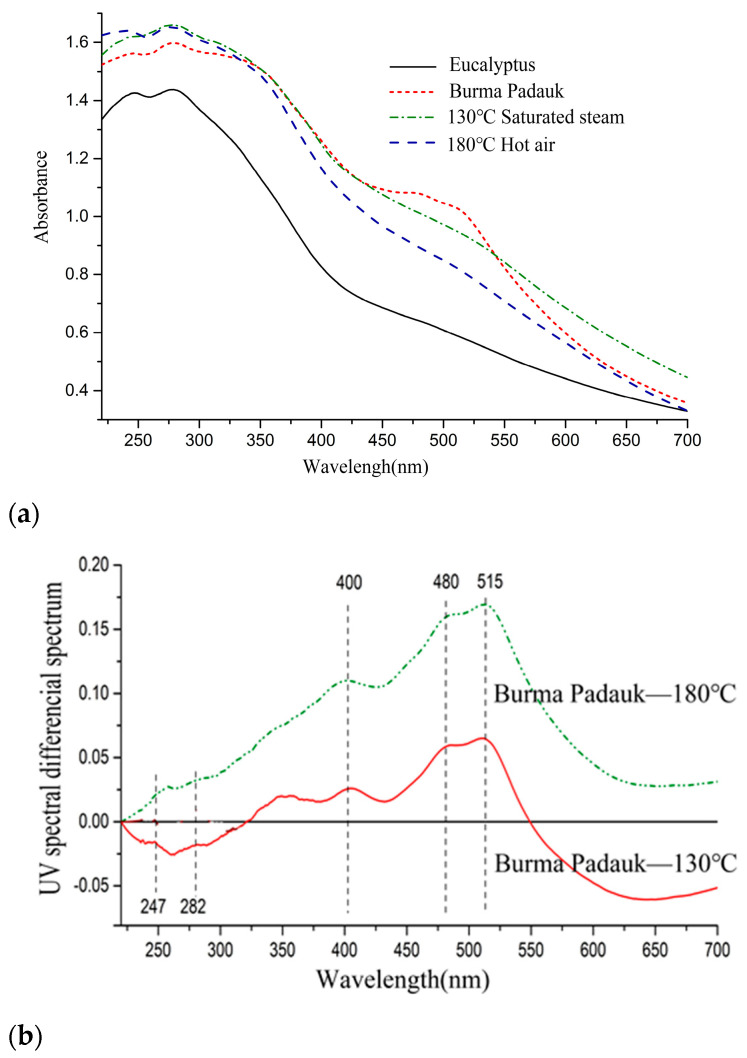
(**a**) UV–Vis spectra and (**b**) UV–Vis differential spectra of Eucalyptus and Burma padauk wood.

**Figure 3 polymers-15-01599-f003:**
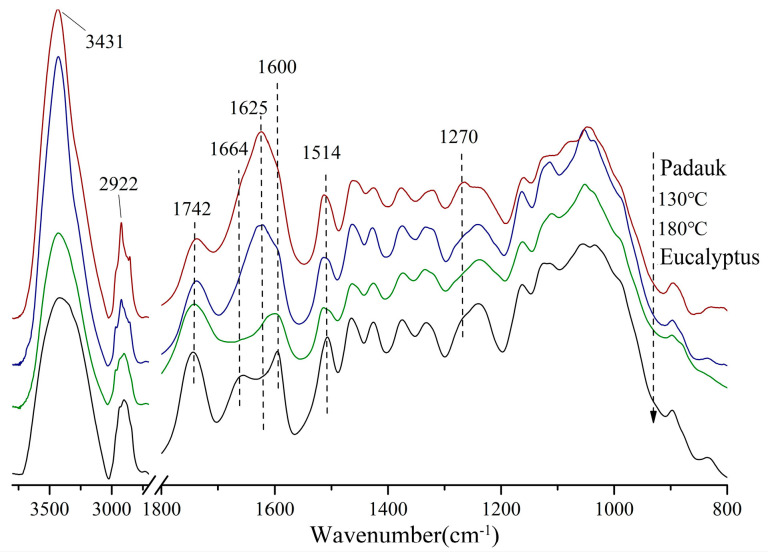
FTIR spectra of Eucalyptus and Burma padauk wood. Burma padauk (red line); 130 °C saturated steam treatment (blue line); 180 °C hot air treatment (green line); control sample Eucalyptus (black line).

**Figure 4 polymers-15-01599-f004:**
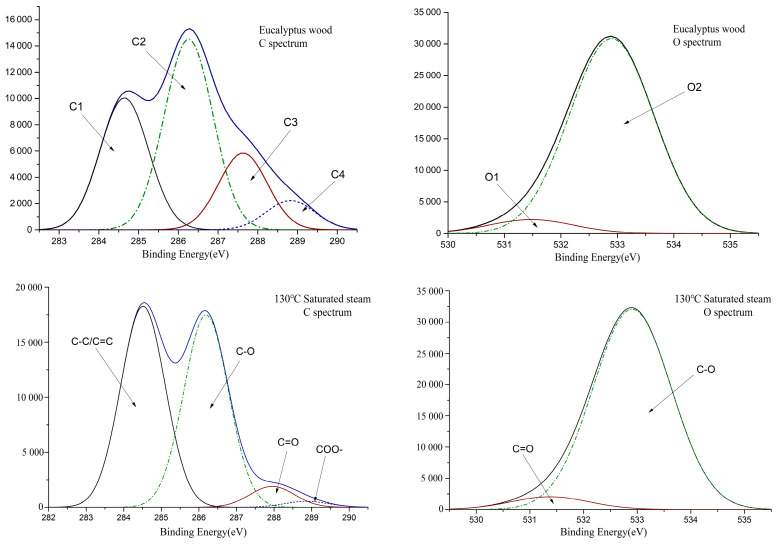
XPS spectra of Eucalyptus wood before and after thermal treatment.

**Figure 5 polymers-15-01599-f005:**
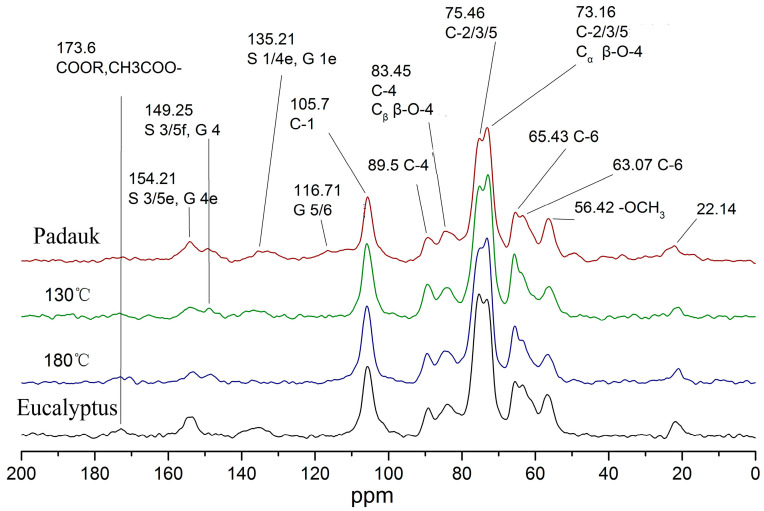
CPMAS-^13^C-NMR spectra of Eucalyptus and Burma padauk wood.

**Table 1 polymers-15-01599-t001:** Change in relative peak height of FTIR for Eucalyptus and Burma padauk wood.

Wavenumber/cm^−1^	3431	1742	1664	1625	1600	1270
Treated Samples	Hydroxyl	Non-Conjugated Carbonyl	Carbonyl Group Conjugated to Aromatic Ring	Conjugated Double Bond	Benzene Ring	Guaiacol
Burma padauk	2.51	0.63	1.01	1.51	1.29	1.33
130 °C	2.84	0.77	0.79	1.29	1.12	1.17
180 °C	1.76	1.04	0.66	0.79	0.94	1.11
Eucalyptus	1.27	0.89	0.71	0.71	0.86	1.09

**Table 2 polymers-15-01599-t002:** Subpeak area fractions of C_1S_ and O_1S_ in XPS spectra.

Treated Samples	C1s Relative Peak Area					O1s Relative Peak Area		
	C1	C2	C3	C4	C3/C2	O1 (531.7 eV)	O2 (532.8 eV)	O1/O2
	(284.7 eV)	(286.2 eV)	(287.6 eV)	(289 eV)				
Burma padauk	66.04	27.62	3.56	2.76	0.13	9.99	90.00	0.11
130 °C	47.66	45.58	4.40	2.34	0.10	8.96	91.03	0.10
180 °C	41.24	45.00	9.45	4.30	0.21	9.26	90.73	0.10
Eucalyptus	31.85	44.37	18.56	5.20	0.42	6.61	93.38	0.07

**Table 3 polymers-15-01599-t003:** NMR data of Eucalyptus and Burma padauk wood.

ppm	Peak Attribution	Relative Number (/C1)			
		Burma Padauk	130	180	Eucalyptus
173.6	COOR, CH3COO-hemicellulose	0.06	0.03	0.08	0.09
154.21	S 3/5 lignin	0.36	0.13	0.13	0.3
149.25	G 4 lignin	0.19	0.07	0.08	——
135.21	S 1/4e, G 1e lignin	0.29	0.13	-0.01	0.17
116.71	G 5/6 lignin	0.39	——	0.05	——
89.5	C-4 crystalline region cellulose	0.34	0.35	0.27	0.24
83.45	C-4 Amorphous Region cellulose; Cβ β-O-4 lignin	0.58	0.48	0.48	0.5
75.46	C-2/C-3/C-5 cellulose Cα β-O-4 lignin	3.39	3	2.97	2.91
65.43	C6 crystalline region cellulose	0.4	0.57	0.46	0.39
63.07	C6 amorphous region cellulose	0.64	0.37	0.41	0.64
56.42	-OCH_3_ lignin	0.1	0.46	0.35	0.51
22.14	Acetyl	0.4	0.1	0.17	0.15

## Data Availability

All the data in this paper will be uploaded.

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
