# Peer review of "Study on the Discoloration Mechanism of Eucalyptus Wood during Thermal Treatment in Different Media"

_polymers, 2023, doi:10.3390/polym15071599_

Round 1
Reviewer 1 Report
General Comments to the Authors
General structure of this work titled “Study on the Wood Discoloration Mechanism of Chromophore Structure under Conventional Thermal Treatment” seems to be appropriate for “Polymers”. The authors investigated some color characteristics of thermally modified Eucalyptus wood samples and Burma Padauk wood samples. This is an interesting study. However, the authors should enhance quality of the manuscript before publication.
Specific comments are as follows:
The manuscript should be revised by a native English speaker.
More literature search should be done on properties of thermal modified wood and summarize in the Introduction section of this manuscript.
It is highly recommended that the authors should read and summarize the below articles into the manuscript.
· Gonultas, O., Candan, Z. 2018. Chemical characterization and FTIR spectroscopy of thermally compressed eucalyptus wood panels. Maderas. Ciencia y tecnología 20(3): 431 – 442.
· Dogu, A. D., Tuncer, F.D., Bakir, D., Candan, Z. 2017. Characterizing microscopic changes of paulownia wood under thermal compression. BioResources 12(3): 5279 – 5295.
Abstract should include more scientific data.
Heat treatment procedure should be explained in detail.
Did the authors apply the heat treatment on the wood powders or solid wood samples? All they should be clarified.
How many samples did the authors use for the treatment or tests?
Test methods should be explained in detail.
Photographs of wood materials and experimental set up should be added into the manuscript.
Did the authors obtain microscopic images (SEM)? It should be supplied to explain/support the results.
Some conclusions and suggestions regarding with industrial perspective should be added into the Conclusions section of the manuscript.
Author Response
- The manuscript should be revised by a native English speaker.
Thanks for your guidance. We have invited a native English-speaking researcher to participate in manuscript revision, please review again.
- More literature search should be done on properties of thermal modified wood and summarize in the Introduction section of this manuscript. It is highly recommended that the authors should read and summarize the below articles into the manuscript.:
Gonultas, O., Candan, Z. 2018. Chemical characterization and FTIR spectroscopy of thermally compressed eucalyptus wood panels. Maderas. Ciencia y tecnología 20(3): 431 – 442.
Dogu, A. D., Tuncer, F.D., Bakir, D., Candan, Z. 2017. Characterizing microscopic changes of paulownia wood under thermal compression. BioResources 12(3): 5279 – 5295.
We have added references and the above article has been cited.
- Abstract should include more scientific data.
The scientific data and the mechanism of this study were added in the abstract.
- Heat treatment procedure should be explained in detail.
Thermal treatments were performed in hot air and saturated steam, respectively. 10 samples for each treatment. Each sample was 1.0 g of wood flour spread out in a glass dish. Wood powder was used for heat treatment for the following considerations:(1) to avoid the uneven heating caused by heat transfer during treatment; (2) to meet the re-quirements of instrument testing for test pieces (FTIR, NMR, XPS); (3) to focus on changes in wood chemical structure, ignoring the impact of wood structure.
For hot air treatment, samples were heated in an electric blast drying oven at 180°C for 12 hours. Before heat treatment, the oven should be preheated to the specified temperature. After thermal treatments, the wood powder needs to be placed for 2 hours and measured after the moisture content was stable.
Saturated steam treatment was performed in pressure sterilizer (Binjiang Medical LS-35HD, Jiangyin, China) at 130°C for 12 hours. The pressure was controlled at 0.27 MPa. Before heat treatment, distilled water should be added to ensure the generation of saturated steam. The glass dish with diameter 15 mm was selected to ensure that the wood powder is in contact with the saturated steam as large as possible. Each glass dish was added 3~5g wood powder to be treated. The glass dish was placed on the bottom of the container. After thermal treatments, the wood powder needs to be placed for 2 hours and measured after the moisture content was stable.
- Did the authors apply the heat treatment on the wood powders or solid wood samples? All they should be clarified.
10 samples for each treatment. Each sample was 1.0 g of wood flour spread out in a glass dish. Wood powder was used for heat treatment for the following considerations:(1) to avoid the uneven heating caused by heat transfer during treatment; (2) to meet the re-quirements of instrument testing for test pieces (FTIR, NMR, XPS); (3) to focus on changes in wood chemical structure, ignoring the impact of wood structure.
- How many samples did the authors use for the treatment or tests?
10 samples for each treatment. Each sample was 1.0 g of wood flour spread out in a glass dish.
- Test methods should be explained in detail.
The test methods detail was supplemented in 2.2-2.6.
- Photographs of wood materials and experimental set up should be added into the manuscript.
Due to the obvious color difference caused by light interference during the actual shooting process, the picture cannot clearly show the color difference before and after processing. Therefore, this paper uses the chromaticity index of a high-precision colorimeter to quantify the color change.
- Did the authors obtain microscopic images (SEM)? It should be supplied to explain/support the results.
Thank you very much for your comments. Since this study is to explore the mechanism of discoloration through chemical structure changes. SEM has a limited degree of characterization of the changes in the molecular level of wood caused by heat treatment. And no components were added in the sample. So the SEM characterization didn’t included in the experimental design. We will still take your opinion seriously.
- Some conclusions and suggestions regarding with industrial perspective should be added into the Conclusions section of the manuscript.
Conclusions and suggestions regarding with industrial perspective were added as follows:
This study provided a theoretical reference for wood thermal discoloration. The experiments have proved that the color of fast-growing plantation Eucalyptus wood can be regulated under thermal treatment, and the treated wood color can be close to the color of Burma Padauk. This research explored a new path for wood surface color regulation in the current heat treatment process of panel or veneer. Heat treatment can’t fully meet the goal of color regulation, and it needs to be combined with other chemical reactions to realize the regulation of chromophore system.
Reviewer 2 Report
The topic of the research work and manuscript is really interesting and provides new information. However there are several issues to be addressed towards its quality improvement before publication.
In line 14, the phrase "were Analyze" needs improvement. The key words could be simplified in order the readers to detect the topic accurately. The term "hemicellulose" should be replaced by the plural form "hemicelluloses". In line 41, subtitute with "relative humidity". In line 45-46, you rather call it "relative humidity" and avoid the simultaneous use of words "although" and "but". In line 50, the word "is" is missing. In line 51, the "less" could be sbustituted with "lower amount". 53-54 lines are not of clear meaning. The latin names should be in italics. The introduction section in general is quite poor, needs enrichment through literature review and presentation of a more detailed state-of-the-art analysis. Please, incorporate as well the relevant paper entitled "Effect of thermal treatment on colour and hygroscopic properties of poplar wood".
In any case the number of 23 references used seems to be quite low (while there are plenty of those, qualitative and detailed studies). There are several points where additional spaces have been used. How many trunks of eucalyptus have been used? Please provide as well the standard used for the extraction process. You did not refer in detail to the equipment used for these steps (samples preparation, devices, model, manufacturer, country etc.). In line 69, an "is" is missing. It is not clear if you have been prepared the samples yourself. Which was the age, number etc. of Burma padauk tree species?.The chromatic indexes should be very serious with them. Statistical analysis is not observed or described. Please, at least provide the standard deviation values on the bars of graphs.
Author Response
- In line 14, the phrase "were Analyze" needs improvement.
It has been edited to “was analyzed”
- The key words could be simplified in order the readers to detect the topic accurately.
The key words were simplified in five words include:” chromophore structure; thermal treatment; discoloration mechanism; color regulation”.
- The term "hemicellulose" should be replaced by the plural form "hemicelluloses".
It has been edited to “hemicelluloses”
- In line 41, subtitute with "relative humidity".
It has been edited to “relative humidity”
- In line 45-46, you rather call it "relative humidity" and avoid the simultaneous use of words "although" and "but".
It has been edited to “relative humidity”. The words "although" and "but" were changed.
- In line 50, the word "is" is missing.
The word "is" has been implemented.
- In line 51, the "less" could be substituted with "lower amount".
The sentence include "less" in line 51 has been substituted.
- 53-54 lines are not of clear meaning.
The sentencein line 53-54 has been substituted.
- The latin names should be in italics.
The latin names have been writen in italics.
- The introduction section in general is quite poor, needs enrichment through literature review and presentation of a more detailed state-of-the-art analysis.
Thank you very much for your comments. We have rewritten the introduction and implemented more detailed state-of-the-art analysis.
- Please, incorporate as well the relevant paper entitled "Effect of thermal treatment on colour and hygroscopic properties of poplar wood".
We have added relevant references and the above article has been cited.
- In any case the number of 23 references used seems to be quite low (while there are plenty of those, qualitative and detailed studies).
We have added 42 relevant references in the revised paper.
- There are several points where additional spaces have been used.
The entire text has been checked and similar situations was avoided.
- How many trunks of eucalyptus have been used?
Eucalyptus wood has the universal characteristics of low extractive content in the chemical composition. We take a trunk of 3-5 years old log and break part of it into wood powder. To study the relationship between the change of the chromophore system and the color in the chemical composition to reveal the relevant mechanism. The influence of the structural characteristics of the wood itself on the color of the heat treatment were avoided.
- Please provide as well the standard used for the extraction process.
According to ASTM D1105-1996, the extractives in the powder were removed by ex-traction process with benzene/alcohol (v/v, 2/1) in a Soxhlet extractor for 24 h.
- You did not refer in detail to the equipment used for these steps (samples preparation, devices, model, manufacturer, country etc.).
The equipment detail was implemented in 2.2-2.6.
- In line 69, an "is" is missing.
The word "is" has been implemented.
- It is not clear if you have been prepared the samples yourself. Which was the age, number etc. of Burma padauk tree species?.
The preparation process has beem implemented in 2.1 and 2.2 as follows
“Thermal treatments were performed in hot air and saturated steam, respectively. 10 samples for each treatment. Each sample was 1.0 g of wood flour spread out in a glass dish. wood powder was used for heat treatment for the following considerations:(1) to avoid the uneven heating caused by heat transfer during treatment; (2) to meet the re-quirements of instrument testing for test pieces (FTIR, NMR, XPS); (3) to focus on changes in wood chemical structure, ignoring the impact of wood structure.
For hot air treatment, samples were heated in an electric blast drying oven at 180°C for 12 hours. Before heat treatment, the oven should be preheated to the specified temperature. After thermal treatments, the wood powder needs to be placed for 2 hours and measured after the moisture content was stable.
Saturated steam treatment was performed in pressure sterilizer (Binjiang Medical LS-35HD, Jiangyin, China) at 130°C for 12 hours. The pressure was controlled at 0.27 MPa. Before heat treatment, distilled water should be added to ensure the generation of saturated steam. The glass dish with diameter 15 mm was selected to ensure that the wood powder is in contact with the saturated steam as large as possible. Each glass dish was added 3~5g wood powder to be treated. The glass dish was placed on the bottom of the container. After thermal treatments, the wood powder needs to be placed for 2 hours and measured after the moisture content was stable.”
- The chromatic indexes should be very serious with them. Statistical analysis is not observed or described. Please, at least provide the standard deviation values on the bars of graphs.
Thank you very much for your instruction. The standard deviation values were added in the bars of graphs.
Round 2
Reviewer 2 Report
In figure 1, the standard deviation values have not been added in the bars of the graph. The colour indexes have not presented in italics in the whole text, not even in graphs. There are several grammatical errors, such as " Eucalyptus wood(Eucalyptus grandis urophylla)were collected". In 102 line, please improve "It was harvest". In 303 line, correct "to regulated". The relevant paper entitled "Effect of thermal treatment on colour and hygroscopic properties of poplar wood" was not cited, as the authors refer to their response (No 11). In the last paragraph of introduction the meaning, scope and significance of your work is not adequately highlighted. Which is the necessity of searching the specific changes? In 2.1. chapter, therefore, was it 1 trunk that was used? Leave spaces between the values and units. conclusions chapter provides significant information, though fails to touch the real significance and practical meaning of the findings of this article. Is there any special reason why to present eucalyptus species or burma padauk in capital the first letter?
Author Response
The reviewer's comments and revision instructions
Specific comments are as follows:
- In figure 1, the standard deviation values have not been added in the bars of the graph.
Since this article emphasizes the color change trend and change range caused by different processing methods, the standard deviation of the chromaticity value in the thermal process has no impact on the above trends, and adding information such as standard deviation in the image will form a certain influence for readers to observe the change trend as interference information. So, it is recommended not to add the numerical information of the standard deviation in the figure 1. Whether the experts consider it necessary to add tabular data as supplementary information to this article, please feel free to let us know.
- The colour indexes have not presented in italics in the whole text, not even in graphs.
The colour indexesL*a*b* have been converted in italics in the whole text and the graphs.
- There are several grammatical errors, such as " Eucalyptus wood(Eucalyptus grandis urophylla)were collected".
The correct writing is” Eucalyptus grandis ´ urophylla”
- In 102 line, please improve "It was harvest".
It has corrected in “The tree was 5 years old”
- In 303 line, correct "to regulated".
It has corrected in “to regulate”
- The relevant paper entitled "Effect of thermal treatment on colour and hygroscopic properties of poplar wood" was not cited, as the authors refer to their response (No 11).
The relevant paper has cited as the 7th reference.
- In the last paragraph of introduction the meaning, scope and significance of your work is not adequately highlighted.
The last paragraph of introduction has been modified as follows:
This study intends to select the fast-growing tree species Eucalyptus (Eucalyptus grandis urophylla) as the control species and take the main precious tree species Burma Padauk (Pterocarpus macrocarpus) as the target tree species to compare the color and chromophore system differences after thermal treatment in hot air and saturated steam. By means of FTIR, UV, NMR, XPS, the adjustability of the chromophore system was investigated. The wood color change under different heat treatment conditions (me-dium, temperature, treatment time) was compared to find the best process. The aim of this study is to change the color of Eucalyptus wood(fast-growing species) tends to the color of Burma Padauk (Pterocarpus macrocarpus) , which upgrade the Eucalyptus by change its color, (Precious tree species of red sandalwood with large fruit, thus in-creasing the value of timber.
- Which is the necessity of searching the specific changes?
The purpose of this paper is to explore the correlation between color change and chromophore change, making color a tunable parameter. The basic reason of color change is the change of chemical structure, and exploring the change of chemical structure and its difference under different thermal conditions is a necessary means to achieve color regulation.
- In 2.1. chapter, therefore, was it 1 trunk that was used?
It has been modified as follows:
Wood logs were cut to discs and 3 discs from different height from the trunk were chosen the cut to pieces and grounded into powder
- Leave spaces between the values and units.
Thanks for the tip, the space between the value and the unit has been filled.
- conclusions chapter provides significant information, though fails to touch the real significance and practical meaning of the findings of this article.
The following importance and practical implications have been added to the conclusion
The following importance and practical implications have been added to the conclusion:
Under different heat treatment conditions, both pressure and temperature were important heat treatment conditions that aggravate the color change. Through the color change caused by the chemical structure change, it is proved that the color was a tunable wood parameter under the condition of high temperature and high pressure thermal treatment.
- Is there any special reason why to present eucalyptus species or burma padauk in capital the first letter?
Uppercase initial letter has been changed to lowercase.
